# Analysis of the Bacterial Community and Fatty Acid Composition in the Bacteriome of the Lac Insect *Llaveia axin axin*

**DOI:** 10.3390/microorganisms13081930

**Published:** 2025-08-18

**Authors:** Reiner Rincón-Rosales, Miriam Díaz-Hernández, Luis Alberto Manzano-Gómez, Francisco Alexander Rincón-Molina, Víctor Manuel Ruíz-Valdiviezo, Adriana Gen-Jiménez, Juan José Villalobos-Maldonado, Julio César Maldonado-Gómez, Clara Ivette Rincón-Molina

**Affiliations:** 1Laboratorio de Ecología Genómica, Tecnológico Nacional de México, Instituto Tecnológico de Tuxtla Gutiérrez, Tuxtla Gutiérrez C.P. 29050, Chiapas, Mexico; reiner.rr@tuxtla.tecnm.mx (R.R.-R.); dihmiriam@gmail.com (M.D.-H.); contacto@3rbiotec.com (L.A.M.-G.); francisco.rm@tuxtla.tecnm.mx (F.A.R.-M.); victor.rv@tuxtla.tecnm.mx (V.M.R.-V.); d10270415@tuxtla.tecnm.mx (A.G.-J.); juan.vm@tuxtla.tecnm.mx (J.J.V.-M.); d07270254@tuxtla.tecnm.mx (J.C.M.-G.); 2Departamento de Investigación y Desarrollo, 3R Biotec SA de CV, Tuxtla Gutiérrez C.P. 29000, Chiapas, Mexico

**Keywords:** bacterial community, fatty acid profile, lac insect, *Llaveia axin axin*, metagenomics, symbiotic bacteria

## Abstract

Microbial symbioses play crucial roles in insect physiology, contributing to nutrition, detoxification, and metabolic adaptations. However, the microbial communities associated with the lac insect *Llaveia axin axin*, an economically significant species used in traditional lacquer production, remain poorly characterized. In this study, the bacterial diversity and community structure of *L. axin axin* were investigated using both culture-dependent and culture-independent (metagenomic) approaches, combined with fatty acid profile analysis. The insects were bred at the laboratory level, in controlled conditions, encompassing stages from eggs to adult females. Bacterial strains were isolated from bacteriomes and identified through 16S rRNA gene amplification and genomic fingerprinting through ARDRA analysis. Metagenomic DNA was sequenced using the Illumina MiSeq platform, and fatty acid profiles were determined by gas chromatography–mass spectrometry (GC-MS). A total of 20 bacterial strains were isolated, with *Acinetobacter*, *Moraxella*, *Pseudomonas*, and *Staphylococcus* detected in first-instar nymphs; *Methylobacterium*, *Microbacterium*, and *Bacillus* in pre-adult females; and *Bacillus* and *Microbacterium* in adults. Metagenomic analysis revealed key genera including *Sodalis*, *Blattabacterium*, and *Candidatus Walczuchella*, with *Sodalis* being predominant in early stages and Blattabacteriaceae in adults. Fatty acid analysis identified palmitic, oleic, linoleic, arachidic, and stearic acids, with stearic acid being the most abundant. These results suggest that dominant bacteria contribute to lipid biosynthesis and metabolic development in *L. axin axin*.

## 1. Introduction

Microbial symbioses are critical for insect physiology, supporting a range of functions such as nutrition acquisition, detoxification, immune defense, and the adaptation to specialized diets. Many insects, particularly sap-feeders, depend on bacterial endosymbionts for the synthesis of essential amino acids, vitamins, and lipids, compensating for the nutritional deficiencies of their diets [1,2]. These symbionts have also been shown to influence insect reproduction, development, thermal tolerance, and protection against pathogens [3,4].

Bacterial endosymbionts not only supplement amino acid and vitamin deficiencies but also contribute to lipid metabolism in their insect hosts. Recent studies have highlighted that symbionts can influence pathways involved in fatty acid and phospholipid biosynthesis, which are critical for membrane formation and energy storage in insects [5]. For instance, in the viviparous tsetse fly, *Glossina morsitans morsitans*, the elimination of obligate symbionts severely impairs lipid metabolism via disruption of the Kennedy pathway, resulting in defective lipid biosynthesis and compromised reproduction [6].

The lac insect *Llaveia axin axin* (Coccoidea: Monophlebidae) is traditionally used by indigenous communities in Mexico and Guatemala for producing lacquered crafts [7]. Females secrete a lipid-rich resin known as “Axe”, utilized for decoration and in traditional medicine to treat burns and skin infections [8]. Early developmental stages of *L. axin axin* occur on young leaves and stems of host plants such as *Acacia cochliacantha*, *Acaciella angustissima*, *Spondias* sp., and *Jatropha curcas* [9]. These host plants produce tannins, toxic compounds that suggest a key role of the insect’s endosymbiotic bacteria in detoxification processes [10].

Insect endosymbionts typically inhabit specialized cells called bacteriocytes, organized into bacteriomes [11]. Approximately 15% of insect species maintain such associations, and the bacteria involved often present highly reduced genomes focused on providing essential functions to their hosts [12].

Symbiotic associations between insects and microorganisms play essential roles in the host’s nutrition, development, and ecological adaptation. Among *Hemiptera*, endosymbionts are particularly important in sap-feeding lineages, where symbiotic bacteria complement deficient diets by synthesizing essential nutrients. In members of the superfamily *Coccoidea* (scale insects), symbiotic systems are diverse and often include specialized bacteriomes housing obligate endosymbionts. Within this group, the Monophlebidae family is known to host complex microbial communities, including both obligate and facultative symbionts. One such species is *Llaveia axin axin*, a phloem-feeding giant scale insect endemic to Mesoamerica. Previous studies have shown that this insect harbors two main bacterial symbionts: *Walczuchella monophlebidarum* (Flavobacteriaceae), an obligate intracellular bacterium residing in the bacteriocytes, and *Enterobacteriaceae* species, which are potentially involved in nitrogen recycling and vitamin synthesis [13].

However, the full extent of bacterial diversity associated with *L. axin axin* across its developmental stages remains unexplored. In this study, we employed high-throughput 16S rRNA gene sequencing to characterize the structure and composition of bacterial communities inhabiting the bacteriomes of *L. axin axin* at different stages of development, expanding our understanding of its microbiome dynamics [13].

Genome-scale metabolic models of several insect endosymbionts reveal that despite genome reduction, these bacteria retain biosynthetic pathways that complement host metabolism, including contributions to fatty acid and phospholipid metabolism [14]. Such metabolic complementarity is evident in sap-feeding insects, where collaborative pathways between host and symbionts ensure the production of essential lipids and membrane components [15]. This co-evolutionary integration allows insects to thrive on nutritionally unbalanced diets and may be critical in species such as *L. axin axin*, which synthesizes large amounts of resin rich in lipids.

Next-generation sequencing (NGS) approaches have enabled detailed studies of insect microbiomes, revealing the metabolic contributions of endosymbionts to their hosts [16]. For instance, in the cochineal insect *Dactylopius coccus*, symbionts contribute to amino acid biosynthesis and nitrogen metabolism [17]. Moreover, metabolomic analyses, including fatty acid profiling, have shown that insect-associated microbiota can influence lipid metabolism, membrane biogenesis, and energy storage [5,18]. Some symbionts, such as *Sodalis* and *Candidatus Walczuchella*, have been implicated in pathways crucial for host survival, including lipid biosynthesis [13].

Proteomic and transcriptomic analyses of insect–symbiont systems have further identified high levels of enzymes involved in fatty acid metabolism within endosymbionts. For example, in *Camponotus chromaiodes* ants, the endosymbiont *Blochmannia* expresses enzymes involved in the biosynthesis of essential fatty acids and membrane lipids, supporting both the nutritional and physiological needs of the host [19]. Similar observations in *Blattella germanica* indicate that symbiont loss leads to marked alterations in host fat body metabolism, underscoring the symbiont’s role in lipid homeostasis [20].

Given that lipids are major components of the lacquer produced by *L. axin axin*, understanding the microbial contributions to fatty acid biosynthesis could clarify the physiological processes underlying resin production and insect development. However, despite advances in symbiosis research, the microbial communities associated with *L. axin axin* and their metabolic roles remain poorly characterized.

Based on this context, we hypothesize that the bacterial symbionts inhabiting the bacteriome of *L. axin axin* significantly contribute to the insect’s fatty acid biosynthesis and overall metabolic homeostasis throughout its development.

Therefore, the aim of this study was to investigate the structure and diversity of the symbiotic bacterial community associated with *L. axin axin* across different developmental stages, using both culturable and metagenomic approaches, alongside the characterization of fatty acid profiles. These findings will provide insights into the metabolic interactions between *L. axin axin* and its microbiota, enhancing our understanding of their role in lacquer biosynthesis and informing future conservation strategies for this economically and culturally important insect.

## 2. Materials and Methods

### 2.1. Neuro-Fuzzy Controlled Biosystem for Artificial Breeding of L. axin axin

A neuro-fuzzy controlled biosystem was implemented to rear *Llaveia axin axin* under stable laboratory conditions (Figure 1). To simulate the insect’s natural habitat, the system integrated neural network training with fuzzy logic-based real-time regulation, adjusting environmental parameters such as temperature, relative humidity, and soil moisture in response to developmental stage. *Jatropha curcas* seedlings were cultivated and acclimated for seven days before carefully placing freshly laid eggs of *L. axin axin* at the base of each plant. The complete life cycle, from hatching to adult emergence, was monitored under these dynamic conditions. Specimens were collected at key timepoints for genomic and biochemical analyses. Individuals designated as “pre-adult females” corresponded to the third-instar stage, occurring approximately 90–100 days post-hatching, just before attaining full maturity. Identification was based on morphological traits such as increased body size, partial wax secretion, and transition to sessile behavior on host stems, in agreement with the ontogenetic stages depicted [21].

### 2.2. Sample Processing and Bacteriome Extraction in the Lac Insect L. axin axin

Samples were collected from different developmental stages of *L. axin axin*, including the egg (E), first-instar nymph (FIN), pre-adult female (PAF), and adult female (AF) stages. Each specimen was carefully cleaned to remove the white wax accumulated on the cuticle, followed by surface disinfection through immersion in 70% ethanol for 10 min, and subsequently washed multiple times with sterile phosphate-buffered saline (PBS) to eliminate any residual contaminants. Under sterile conditions, individual *L. axin axin* females were dissected using fine sterile forceps under a stereoscopic microscope to extract the targeted tissues. Bacteriomes were identified as paired, ovoid, whitish organs located in the posterior abdominal cavity, positioned laterally to the ovaries and ventrally to the hindgut (Figure 2). These structures were clearly distinct from surrounding tissues due to their color, texture, and bilateral symmetry. Some samples were immediately processed and analyzed after extraction, while others were preserved for subsequent analyses. To preserve sample integrity and prevent oxidative degradation, the extracted materials were thoroughly washed with sterile phosphate-buffered saline (PBS) and subsequently stored in 70% ethanol, following the standardized protocol established by Ramírez-Puebla et al. [17].

### 2.3. Culture-Independent Characterization of Bacteria Associated with the Bacteriomes of L. axin axin

Metagenomic DNA was extracted from the bacteriomes of *L. axin axin* at different developmental stages, excluding eggs, using the Qiagen DNeasy^®^ Blood & Tissue Kit (Qiagen, Hilden, Germany) following the manufacturer’s protocol [13]. Approximately 20–40 females were surface-sterilized, dissected under a stereoscopic microscope, and bacteriomes were isolated with sterile fine forceps and preserved in sterile 0.85% NaCl solution. Pooled samples were weighed (25 mg for first-instar nymphs, pre-adult females, and adult females) to optimize cell lysis. DNA extraction was carried out by overnight incubation at 56 °C with Buffer ATL and proteinase K, followed by sequential washes with Buffers AW1 and AW2, and elution in Buffer AE (Qiagen, Hilden, Germany). DNA integrity and purity were evaluated using a NanoDrop UV-Vis spectrophotometer (Thermo Fisher Scientific, Waltham, MA, USA), with an A260/A280 ratio > 1.8 considered acceptable. The V3–V4 region of the 16S rRNA gene was amplified using primers Bakt_341F (5′-CCTACGGGNGGCWGCAG-3′) and Bakt_805R (5′-GACTACHVGGGTATCTAATCC-3′) in a PCR thermocycler (Applied Biosystems, Thermo Fisher Scientific, CA, USA) [22]. PCR products were verified by 1% agarose gel electrophoresis stained with SYBR Green and visualized under UV light [23]. Metagenomic DNA and PCR products were purified using the DNA Clean & Concentrator^®^-25 Kit (Zymo Research, Irvine, CA, USA) prior to sequencing. Sequencing was conducted on an Illumina MiSeq 300PE platform (Macrogen, Seoul, Republic of Korea) with 2 × 300 paired-end reads [24]. Data processing in QIIME 2 involved quality filtering (quality score < 25, homopolymers > 6, sequences < 400 bp, and primer or barcode errors). OTUs were clustered at 97% similarity using UCLUST v11 (Edgar, 2010) [25], and chimeric sequences were removed with ChimeraSlayer v 20110519 [26]. Sequences were aligned to the Greengenes database using PyNAST v1.2.2 at a 75% threshold [27], and taxonomic assignments were performed using the RDP classifier with an 80% confidence threshold [28]. Community structure visualization, including heat maps and diversity indices, was completed using RStudio v3.6.2 [29]. Raw sequencing data are available in the Sequence Read Archive (SRA) under BioProject accession number PRJNA1288045.

### 2.4. Genomic Characterization of Symbiotic Bacteria in the Lac Insect L. axin axin

Bacteriomes from first-instar nymphs, pre-adult females, and adult females of *L. axin axin* were dissected under a stereoscopic microscope and pooled into 1.5 mL tubes containing 500 µL of sterile 0.85% NaCl solution. Approximately 0.50 g of bacteriome tissue was used to promote bacterial proliferation. Bacterial suspensions were inoculated into test tubes containing Peptone–Yeast–Calcium (PY-Ca) medium (Peptone 15.0 g, Yeast Extract 1.0 g, CaCl_2_ 5.0 g) and Yeast Extract Mannitol (YEM) medium (Yeast Extract 1.0 g, Mannitol 10.0 g, K_2_HPO_4_ 0.50 g, MgSO_4_·7H_2_O 0.20 g, NaCl 0.10 g, CaCO_3_ 1.0 g), following established protocols [30,31]. Solid media were prepared by supplementing PY-Ca and YEM with 15.0 g of agar. Cultures were incubated at 28 °C for 72 h. After confirming purity, isolates were preserved in 70% glycerol and stored at −20 °C. Morphological characterization was performed by Gram staining and optical microscopy, and colony morphology was assessed according to Bergey’s Manual of Systematic Bacteriology. Total genomic DNA was subsequently extracted exclusively from bacteriomes (excluding eggs) using the ZR Fungal/Bacterial DNA MiniPrep^®^ Kit (Zymo Research, USA), according to the manufacturer’s protocol. DNA integrity was verified by 1% agarose gel electrophoresis, and quantification was performed using a NanoDrop 2000c spectrophotometer (Thermo Fisher Scientific^®^, USA). Genomic fingerprinting was performed using Enterobacterial Repetitive Intergenic Consensus (ERIC)-PCR with primers ERIC1 and ERIC2 [32]. Amplification of the 16S rRNA gene was carried out using universal bacterial primers FD1 (5′-AGAGTTTGATCCTGGCTCAG-3′) and 1492R (5′-AAGGAGGTGATCCAGCC-3′) [33,34], followed by purification of PCR products using the Roche^®^ PCR product purification system (Roche Diagnostics GmbH, Mannheim, Germany). The amplified 16S rRNA products were digested with the restriction enzyme *Hinf* I (5′-G^ANTC-3′) for ARDRA analysis and visualized on 1.5% agarose gels [35]. Bacterial species diversity and abundance were evaluated using the Shannon–Weaver index [36]. Purified 16S rRNA gene products were sequenced by Macrogen^®^ using both forward and reverse primers. Sequences were aligned and edited with BioEdit v7.1.3., and taxonomic identification was performed using BLAST v2.13.0 (NCBI) and the Ribosomal Database Project (RDP) v11 [37]. Phylogenetic analyses were performed using the Neighbor-Joining method with the Tamura–Nei model implemented in MEGA version 10.0 [38]. Sequence submissions to GenBank were carried out using Sequin version 15.10 to obtain accession numbers. All sequences isolated from *L. axin axin* were deposited in GenBank under accession numbers MH223592.1–MH223595.1 and PV888580–PV888584.

### 2.5. Fatty Acid Profile of the Bacteriome in the Lac Insect L. axin axin

Total lipid extraction and fatty acid methyl ester (FAME) preparation were performed to analyze the fatty acid profile of the bacteriome in *L. axin axin*. For lipid extraction, 0.20 g of eggs, 0.30 g of first-instar nymph bacteriomes, 0.30 g of pre-adult female bacteriomes, and 0.30 g of adult female bacteriomes were homogenized in a 2:1 chloroform–methanol mixture cooled on ice. Water was subsequently added to adjust the solvent ratio to 8:4:3 (chloroform–methanol–water), and the samples were sonicated for 10 min in an ice bath. The homogenate was then centrifuged at 5000 rpm for 2 min, resulting in two distinct phases. The lower organic phase, containing the lipid extract, was carefully recovered using a micropipette and transferred to a new tube, followed by the addition of chloroform. All lipid extracts were pooled into clean vials, and this procedure was performed in triplicate [39]. To generate FAMEs, 100 µL of lipid extract was mixed with 1 mL of 2 M NaOH in methanol and heated at 80 °C for 20 min. After cooling, 1 mL of 14% boron trifluoride (BF_3_) in methanol was added, and the mixture was heated again at 80 °C for another 20 min. Methyl esters were extracted using 1 mL of HPLC-grade hexane [40]. Fatty acid analysis was conducted by gas chromatography–mass spectrometry (GC–MS) using an Agilent Technologies 7890A instrument (Agilent Technologies, Santa Clara, CA, USA) equipped with a DB-WAXter column (J&W Scientific 122-7362; 60 m × 250 μm × 0.25 μm). Helium was used as the carrier gas at a flow rate of 1 mL/min, and 1 μL of FAME extract was injected in splitless mode. The injector and detector temperatures were set at 250 °C and 230 °C, respectively. The oven temperature program consisted of an initial hold at 150 °C for 5 min, a ramp of 30 °C/min to 210 °C, an increase of 1 °C/min to 213 °C, and a final ramp of 20 °C/min to 225 °C, where it was maintained for 12 min. Detection was performed using a mass-selective detector (Agilent Technologies 5975C) [40,41].

## 3. Results

### 3.1. Artificial Breeding of L. axin axin Using a Neuro-Fuzzy Controlled Biosystem

The neuro-fuzzy controlled biosystem successfully enabled the complete breeding of *L. axin axin* under laboratory conditions, allowing individuals to progress through all developmental stages: eggs, first-instar nymphs, second- and third-instar nymphs, pre-adult females, and adult females. The specific environmental conditions established for each stage were as follows: 35 °C and 43% relative humidity for eggs; 30 °C and 43% humidity for first-instar nymphs; 23 °C and 30% humidity for pre-adult females; and 28 °C and 35% humidity for adult females. The entire life cycle was completed in approximately 120 days.

Eggs displayed a bright orange coloration and were fully enveloped by a protective waxy layer. First-instar nymphs exhibited fine wax coatings, prominent antennae, visible feeding stylets, and numerous setae. During the second- and third-instar stages, insects developed lighter orange coloration and small protuberances on the exoskeleton, progressing toward the pre-adult stage. Pre-adult females showed an intense orange coloration, a smooth and glossy exoskeleton with small folds, and the emergence of a fine cottony layer. Adult females grew up to 2.5 cm in length, exceeded 0.5 g in weight, developed a dense cottony covering, and became fully sessile on the host plant (Figure 3).

The morphological traits were consistent across replicates, and no abnormalities in growth or molting were observed. The precise environmental control achieved through the neuro-fuzzy system enabled stable developmental performance throughout all trials. This artificial biosystem thus provides valuable biological data and represents a robust platform for advancing microbiological, biochemical, and conservation research on this economically important insect.

### 3.2. 16S rRNA Gene-Based Profiling of Bacterial Communities Associated with L. axin axin

High-throughput sequencing of 16S rRNA gene amplicons obtained from total DNA extracted from the bacteriomes of *L. axin axin* across four developmental stages revealed substantial variation in bacterial community structure and composition. There were a total of 27,847 high-quality reads for first-instar nymphs, 149,875 for pre-adult females, and 174,001 for adult females. After quality filtering, 39,185 and 323,876 high-quality reads were retained for the early and late stages, respectively.

In relation to alpha diversity, metrics were calculated using the Shannon–Weaver index (Table 1). The highest diversity (H′ = 2.0202) was recorded in pre-adult females, followed by first-instar nymphs (H′ = 1.8842). Adult females displayed the lowest diversity (H′ = 0.8101), indicating reduced evenness and the dominance of a few taxa. Species richness (*d*) was highest in first-instar nymphs (*d* = 7.2304), followed by pre-adult females (*d* = 6.2093). A pronounced reduction in richness was observed in adult females (*d* = 1.5745).

Taxonomic classification at the phylum level (Figure 4) revealed a marked ontogenetic restructuring of the bacterial community associated with *Llaveia axin axin*. The nymphal stage exhibited the highest phylum level diversity, with detectable proportions of Verrucomicrobiota, Actinomycetota, Acidobacteriota, Bacteroidota, Chloroflexota, and Pseudomonadota. Among these, Pseudomonadota and Acidobacteriota were notably abundant, suggesting a complex and metabolically versatile microbial community during this early stage.

In contrast, pre-adult and adult females showed a striking simplification of the bacteriome, with near-complete dominance by Bacteroidota. This transition may reflect developmental constraints or selection for specialized endosymbionts optimized for mature host physiology. The consistent dominance of Bacteroidota in later stages underscores their likely importance in the adult insect’s metabolic and symbiotic functioning.

These patterns demonstrate a clear phylum-level succession during development, from a phylogenetically diverse and potentially environmentally acquired microbiota in nymphs, toward a highly specialized and reduced symbiotic community in adult stages.

16S rRNA gene-based metagenomic profiling revealed distinct shifts in the relative abundance and complexity of bacterial genera across the developmental stages of *Llaveia axin axin*. In first-instar nymphs, a highly diverse bacterial community was detected, including notable genera such as *Bradyrhizobium* (Pseudomonadota), *Burkholderia* (Pseudomonadota), *Koribacter* (Acidobacteriota), *Solibacter* (Acidobacteriota), *Immundisolibacter* (Actinobacteriota), *Walczuchella* (Bacteroidota), and *Sodalis* (Pseudomonadota), along with several unclassified taxa (e.g., FJ479568_g, GU127739_g, PAC000121_g). In pre-adult and adult females, the community structure shifted towards near monodominance by the genus *Sodalis* (Pseudomonadota). This pattern suggests a stage-dependent restructuring and simplification of the bacteriome during maturation.

Collectively, these findings reveal dynamic microbial succession across the life cycle of *L. axin axin*, characterized by the early dominance of *Sodalis*, mid-stage diversification with multiple bacterial genera, and late-stage simplification dominated by *Blattabacteriaceae*. These results highlight profound ontogenetic shifts in microbiome structure, closely linked to the physiological and ecological transitions of the lac insect.

### 3.3. ARDRA-Based Genomic Profiling and Diversity Indices of Cultivable Bacteria

ARDRA (Amplified Ribosomal DNA Restriction Analysis) of 16S rRNA gene fragments digested with *Hinf* I enabled the genotypic profiling of cultivable bacterial isolates from the bacteriomes of three developmental stages of *L. axin axin*: first-instar nymphs, pre-adult females, and adult females. A total of 20 isolates were recovered and classified into nine distinct ARDRA genotypic groups (Table 2). First-instar nymphs yielded the highest number of isolates (*n* = 8), distributed into four ARDRA groups: Group I (NI-01, NI-06), Group II (NI-02, NI-05), Group III (NI-03, NI-04, NI-07), and Group IV (NI-08). Pre-adult females produced six isolates, clustered into three profiles: Group I (PA-02, PA-05), Group II (PA-03, PA-04), and Group III (PA-01). Adult females also yielded six isolates, grouped into two profiles: Group I (AD-01, AD-04) and Group II (AD-02, AD-03, AD-05).

Shannon–Weaver diversity indices indicated that first-instar nymphs harbored the most diverse community (*H*′ = 1.32; *d* = 1.44), reflecting a more balanced and moderately rich bacterial composition at this early stage. Pre-adult females showed a moderate decrease in both diversity and richness (*H*′ = 1.05; *d* = 1.24), whereas adult females exhibited the lowest values (*H*′ = 0.67; *d* = 0.62), suggesting reduced taxonomic complexity and dominance of specific bacterial genotypes in mature stages.

These results reveal that cultivable bacterial communities within the bacteriomes of *L. axin axin* undergo ontogenetic shifts in genotypic structure, with greater richness and diversity in earlier stages that progressively decline with insect development.

### 3.4. Molecular Characterization and Taxonomic Classification of Cultivable Bacteria from L. axin axin

Molecular analysis of 16S rRNA gene sequences from bacterial isolates obtained from the bacteriomes of *L. axin axin* revealed a diverse taxonomic composition across developmental stages (Table 3).

In first-instar nymphs, four bacterial genera were identified based on phylogenetic analysis: *Acinetobacter*, *Moraxella*, *Pseudomonas*, and *Staphylococcus*. Strain NI-01 shared 98% sequence identity with *Staphylococcus warneri*, while NI-02 exhibited 99% identity with *Pseudomonas oryzihabitans*. Strain NI-03 showed 98% similarity to *Acinetobacter pittii*, and NI-04 matched *Moraxella osloensis* with 98% identity. These isolates belong to the phyla Pseudomonadota and Bacillota.

In pre-adult females, three bacterial genera were identified: *Methylobacterium*, *Microbacterium*, and *Bacillus*. Strain PA-01 shared 99% sequence identity with *Methylobacterium hispanicum*, PA-04 showed 90% similarity to *Microbacterium marinilacus*, and PA-05 aligned 97% with *Bacillus* sp. These taxa belong to the phyla Pseudomonadota, Actinobacteriota, and Bacillota.

In adult females, two bacterial genera were recovered. The strain AD-03 exhibited 94% sequence identity with *Bacillus* sp., while AD-04 shared 93% similarity with *Microbacterium marinilacus*. The genus *Bacillus* was the most represented taxon at this stage.

These results demonstrate that the cultivable microbiota of *L. axin axin* bacteriomes exhibits developmental stage-specific patterns of taxonomic diversity, with Gammaproteobacteria predominant in early stages and Bacillota and Actinobacteriota gaining prominence in later stages.

Additionally, the phylogenetic analysis based on 16S rRNA gene sequences corroborated the taxonomic affiliations of the bacterial strains isolated from *L. axin axin*, reinforcing the molecular identification results. The Neighbor-Joining trees constructed using the Tamura–Nei model (Appendix A) revealed that each isolate clustered closely with its corresponding reference strain. In first-instar nymphs, strains NI-01 to NI-04 grouped within well-supported clades corresponding to *Staphylococcus*, *Pseudomonas*, *Acinetobacter*, and *Moraxella*, respectively. In pre-adult females, PA-01, PA-04, and PA-05 aligned phylogenetically with *Methylobacterium*, *Microbacterium*, and *Bacillus*. For adult females, AD-03 and AD-04 formed separate branches within the *Bacillus* and *Microbacterium* lineages, respectively, albeit with lower sequence identity. Overall, the tree topology mirrored the stage-specific bacterial composition and validated the evolutionary relationships inferred from sequence similarity, enhancing confidence in the classification and highlighting developmental shifts in the symbiotic microbiota.

### 3.5. Fatty Acid Composition of L. axin axin Across Developmental Stages and Its Bacteriome

The fatty acid profiling of *L. axin axin* across different developmental stages revealed notable differences in the relative abundance of key saturated and unsaturated fatty acids (Table 4). Stearic acid (C18:0) was consistently the most dominant fatty acid in all life stages, ranging from 31.90% in first-instar nymphs to 49.68% in adult females. Its abundance was similarly high in the Axe wax (40.99%), indicating a strong association between this compound and the waxy secretions of the insect.

In eggs, the second most abundant fatty acids were arachidic acid (C20:0) at 17.97% and linoleic acid (C18:2) at 16.14%, while oleic acid (C18:1) and palmitic acid (C16:0) were moderately present. First-instar nymphs showed elevated levels of palmitic acid (21.78%) and a decreased presence of arachidic acid (5.42%). Pre-adult females had an enriched profile of unsaturated fatty acids, with oleic acid reaching 21.62% and linoleic acid 19.20%. In adult females, stearic acid levels peaked (49.68%), while oleic and arachidic acids also remained substantial at 15.22% and 13.29%, respectively. Minor fatty acids such as behenic acid (C22:0), valeric acid (C5:0), and butyric acid (C4:0) were detected in low concentrations, with behenic acid present in early stages and the Axe wax. Notably, undecanoic acid appeared exclusively in the Axe wax (6.92%), highlighting its unique biochemical composition compared to the insect’s tissues. These results indicate that *L. axin axin* exhibits stage-specific lipid metabolic patterns, with stearic acid being the predominant fatty acid throughout development and in the wax secreted by adult females.

## 4. Discussion

Understanding the diversity, composition, and functional roles of insect-associated microbiota is essential for elucidating how symbiotic interactions influence host development, physiology, and ecological adaptation. In scale insects such as *Llaveia axin axin*, the bacteriome represents a specialized niche where complex microbial communities, comprising both obligate and facultative symbionts, can play critical roles in nutrition, development, and the production of ecologically relevant compounds. Our findings contribute to this understanding by combining culture-dependent and culture-independent approaches to describe the dynamics, composition, and potential functional significance of the *L. axin axin* microbiome across developmental stages.

The bacteria isolated on PY-Ca and YEM media do not necessarily represent obligate intracellular endosymbionts, which are typically unculturable under standard aerobic conditions due to their strict dependence on host cells and reduced genomes. Rather, our isolates likely correspond to facultative or transiently associated bacteria, some of which may be extracellular symbionts or environmental microbes colonizing the bacteriome region.

In other insect systems, culturable facultative symbionts have been recovered from tissues associated with bacteriomes or hemolymph. For instance, *Serratia symbiotica* and *Hamiltonella defensa*, both facultative symbionts of aphids, have been successfully cultured on cell-free media under specific conditions [42]. Additionally, studies in stinkbugs have shown that culturable *Burkholderia* spp. symbionts colonize specialized midgut crypts and can be transmitted environmentally [43]. Thus, while most obligate endosymbionts of bacteriocytes (e.g., *Buchnera aphidicola*, *Carsonella* ruddii) are indeed unculturable [44], the bacterial communities associated with the bacteriome may include culturable, aerobic microorganisms of potential ecological relevance.

Moreover, the phylogenetic analysis revealed a pronounced ontogenetic shift in the composition of the bacteriome across developmental stages. Genera such as *Bradyrhizobium* and *Burkholderia* were prominent in nymphs, whereas *Koribacter* and *Solibacter* dominated the bacteriome in pre-adult and adult females. Similar developmental shifts in symbiotic communities have been observed in other hemipterans, including mealybugs and aphids, where early colonization by Gammaproteobacteria gives way to more specialized or diverse bacterial consortia as development progresses [45,46]. The dramatic reduction in microbial diversity observed in adult females suggests selective maintenance of specific endosymbionts, potentially linked to their sessile behavior and involvement in lac secretion [47].

Our results revealed a clear dominance of a Flavobacteriaceae taxon, particularly evident in adult stages (Figure 5). Based on previous research, this bacterium has been identified as *Walczuchella monophlebidarum*, an obligate endosymbiont of *Llaveia axin axin*, localized within specialized bacteriocytes and vertically transmitted across generations [13]. This supports the interpretation that *W. monophlebidarum* is the primary symbiotic partner in this species, consistent with the dominant OTU observed in our phylogenetic data.

The secondary symbiont, previously characterized as a member of the Enterobacteriaceae, was also detected across developmental stages but at a much lower abundance. This pattern is consistent with its proposed facultative role in nutritional complementation or metabolic flexibility, as suggested in other scale insects [13].

By confirming the presence and dominance of *W. monophlebidarum*, our data validate and expand on existing knowledge of the *L. axin axin* microbiome. Moreover, the stage-dependent fluctuation in community structure suggests the dynamic regulation of bacterial symbionts during host development, a pattern observed in other insect–symbiont systems as well.

At a finer taxonomic resolution, the genus-level analysis indicated early dominance of *Sodalis*, an endosymbiont frequently associated with nutritional functions, highlighting its critical role in early host development. Its subsequent decline and replacement by Blattabacteriaceae in adult females mirror the patterns observed in other insects where the host–symbiont composition changes across developmental stages [48,49,50]. For example, in psyllids (Hemiptera: Psylloidea), a shift was observed from *Sodalis*-like bacteria during juvenile stages to *Arsenophonus* or other *Enterobacteriaceae* in adults [51]. Similarly, in weevils of the family Dryophthoridae, *Nardonella* has been replaced in some lineages by Gammaproteobacteria such as *Sodalis* or *Enterobacter* across host evolution, with evidence suggesting possible ontogenetic transitions in bacterial load and function [52,53]. These studies provide clear evidence of taxon-specific shifts in endosymbiont identity linked to developmental or evolutionary dynamics.

The ARDRA-based profiling of cultivable bacteria further corroborated these findings, showing higher genotypic richness in early instars and a progressive decrease in diversity with maturation. This trend suggests that only specific bacterial genotypes are retained into adulthood, likely those essential for adult physiological functions such as wax production or host–plant interactions [54].

Furthermore, the phylogenetic analysis of 16S rRNA sequences confirmed that early stages harbor a broader diversity of genera, including *Acinetobacter*, *Moraxella*, and *Pseudomonas*. These genera are recognized for their metabolic versatility, including involvement in fatty acid metabolism and hydrocarbon degradation, which may aid in cuticular development or the digestion of plant-derived compounds [55]. Collectively, these two approaches offered complementary insights: ARDRA enabled the detection of genotypic patterns among cultivable isolates, while high-throughput 16S profiling captured the full taxonomic structure of the bacterial community, including non-culturable taxa. This integration reinforced the observation of a progressive narrowing of microbial diversity through development.

With respect to lipid metabolism, the fatty acid profile across developmental stages highlighted stearic acid (C18:0) as the most abundant fatty acid in all insect stages and in Axe wax. This consistent presence suggests a potential dual role as both a structural and signaling lipid. In contrast, the relative abundance of other fatty acids, such as palmitic (C16:0), linoleic (C18:2), and arachidic acid (C20:0), varied across life stages, indicating developmental stage-specific lipid profiles. Similar patterns have been observed in the white wax scale insect *Ericerus pela*, where genes related to fatty acid elongation and reduction (e.g., ELOVL, FAR) are associated with wax synthesis [56,57].

Notably, the detection of fatty acids such as behenic and undecanoic acid specifically associated with Axe wax supports the hypothesis that bacterial metabolic contributions may influence wax secretion and potentially contribute to cuticle-related processes. Comparable findings have been reported in other wax-producing insects, where bacterial symbionts participate in the biosynthesis or structural modification of cuticular hydrocarbons and esters [58,59]. Comparable findings have been reported in other wax-producing insects, such as the honeybee (*Apis mellifera*), where bacterial symbionts have been associated with modifications in the composition of cuticular waxes, primarily hydrocarbons and wax esters [60]. In the case of *Ceratitis capitata*, gut bacterial symbionts influence cuticular hydrocarbon profiles that are critical for sexual behavior and potentially for hydrocarbon biosynthesis [61]. Additionally, in leafcutter ants (*Atta sexdens*, *Atta cephalotes*), specific cuticular compounds such as alkyl amides are present only when mutualistic bacteria are associated with the cuticle, suggesting a potential biosynthetic contribution from the symbionts [62]. These cases underscore the potential role of microbial partners in shaping the composition of wax-related compounds in insects.

Taken together, these findings provide compelling evidence that the microbiota of *L. axin axin* is dynamically restructured throughout its life cycle. The integration of microbial community data, cultivable bacterial genotypes, phylogenetic classifications, and fatty acid profiles underscores the vital roles that endosymbionts play not only in the nutritional and developmental biology of the host but also in its capacity to synthesize ecologically and economically important compounds such as Axe wax.

At the functional level, our results allow us to propose plausible symbiotic roles for the major bacterial taxa detected in *Llaveia axin axin*. For example, the early dominance of *Sodalis* suggests its involvement in nutritional provisioning, consistent with its role as a facultative endosymbiont in other *Hemiptera* and *Coleoptera*, where it contributes to the biosynthesis of essential amino acids and cofactors during critical developmental stages [63].

Consistent with previous reports, *Walczuchella monophlebidarum*, described by Rosas-Pérez et al. [13] as the primary symbiont of *L. axin axin*, was consistently detected throughout all developmental stages and was particularly dominant in adult females. This supports its status as an obligate endosymbiont with a highly reduced genome, specialized in essential amino acid biosynthesis, a typical feature of primary symbionts in scale insects.

In addition, the detection of *Blattabacteriaceae* and *Enterobacteriaceae* in later developmental stages suggests potential roles in nitrogen recycling, B-vitamin biosynthesis, and hydrocarbon metabolism. These functions are well-documented in symbionts of cockroaches, aphids, and mealybugs [64,65].

Among the culturable taxa, we recovered genera such as *Pseudomonas*, *Bacillus*, and *Microbacterium*, which are known for their antimicrobial properties, roles in cuticle conditioning, and support of digestive functions in other phytophagous insects. These bacterial genera have also been implicated in plant compound degradation and host immune modulation [66,67].

Together, these findings provide a more comprehensive ecological and functional framework for interpreting the microbial community associated with *L. axin axin*, integrating both culture-dependent and culture-independent data.

## 5. Conclusions

Phylogenetic analysis revealed a dynamic and stage-specific succession of bacterial taxa, with early dominance of *Sodalis* and late-stage enrichment of *Blattabacteriaceae*. ARDRA profiling and 16S rRNA sequencing confirmed the shifts in the cultivable microbiota, while fatty acid profiling identified stearic acid as the predominant compound across all stages and wax extracts. These findings provide key genomic and metabolic insights into symbiont–host interactions in scale insects. This work contributes to the growing field of computational microbial genomics and offers a foundation for advancing the study and sustainable use of beneficial insects in biotechnology, agriculture, and environmental conservation.

## Figures and Tables

**Figure 1 microorganisms-13-01930-f001:**
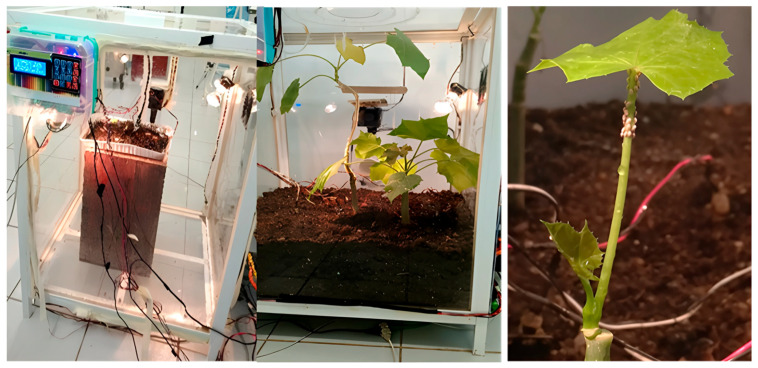
Neuro-fuzzy controlled automated biosystem for the artificial breeding of the lac insect *L. axin axin* under simulated environmental conditions.

**Figure 2 microorganisms-13-01930-f002:**
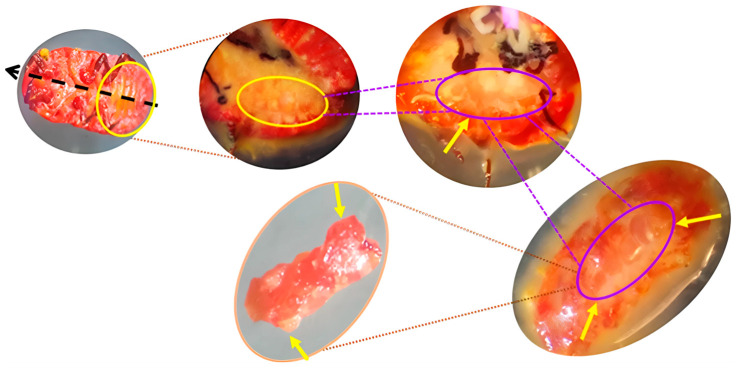
Dissection and identification of the bacteriome in the lac insect *L. axin* axin; yellow arrows indicate the bacteriome within the abdominal region.

**Figure 3 microorganisms-13-01930-f003:**
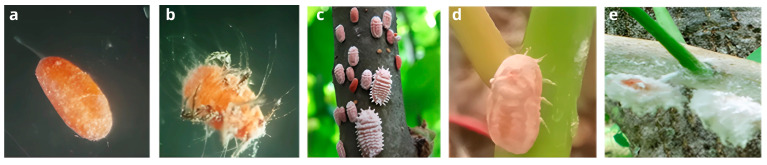
Developmental stages of the lac insect *L. axin axin* observed under controlled breeding conditions: (**a**) egg stage, fully covered by an intense orange waxy coating; (**b**) first-instar nymph with fine wax filaments and visible antennae; (**c**) second- and third-instar nymphs characterized by increased size and denser wax coverage; (**d**) pre-adult female displaying a smooth, glossy exoskeleton and onset of cottony secretion; (**e**) adult female exhibiting a dense cottony covering and complete sessility on the host plant.

**Figure 4 microorganisms-13-01930-f004:**
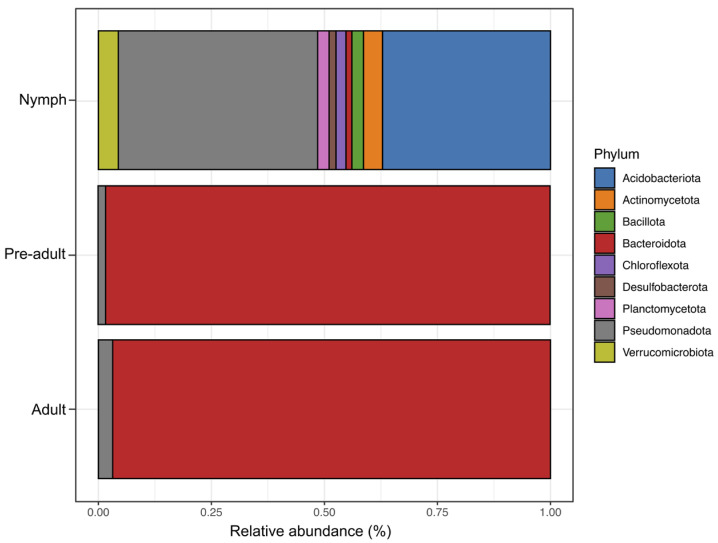
Relative abundance of bacterial phyla across different developmental stages of the lac insect *L. axin axin*.

**Figure 5 microorganisms-13-01930-f005:**
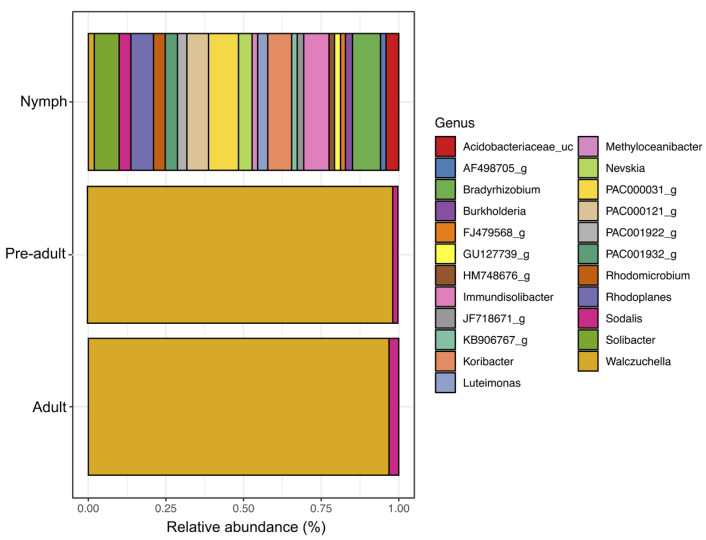
Relative abundance of bacterial genera across four developmental stages of the lac insect *L. axin axin*.

**Table 1 microorganisms-13-01930-t001:** Diversity and abundance indices of bacterial communities across four developmental stages of the lac insect *L. axin axin*.

Developmental Stage	Number of Reads (*n*)	Abundance (d)	Shannon–Weaver Diversity Index (H′)
First-instar nymphs	27,847	7.2304	1.8842
Pre-adult females	149,875	6.2093	2.0202
Adult females	174,001	1.5745	0.8101

**Table 2 microorganisms-13-01930-t002:** Diversity and abundance indices of bacterial species isolated from different developmental stages of the lac insect *L. axin axin*.

Sample Source	No. of Isolates	No. of Groups ARDRA Profiles ^a^	Relative Abundance (%)	Shannon–Weaver Index ^b^	
				Richness (*d*)	Diversity (*H*′)
First-instar nymphs	8	4	40	1.44	1.32
Pre-adult females	6	3	30	1.24	1.05
Adult females	6	2	30	0.62	0.67
Total	20	9	100		

^a^ ARDRA profiles, amplified rRNA restriction analysis obtained with HinfI restriction enzyme. ^b^ Estimated using the Shannon–Weaver Index.

**Table 3 microorganisms-13-01930-t003:** Phylogenetic affiliation of bacterial strains isolated from bacteriomes of *L. axin axin* based on 16S rRNA gene sequences.

Strain	Closest NCBI Match ^a^/Identity (%)	Accession Number	Developmental Stage	Phylum
NI-01	*Staphylococcus warneri* (98%)	MH223592.1	First-instar nymph	Bacillota
NI-02	*Pseudomonas oryzihabitans* (99%)	MH223593.1	First-instar nymph	Pseudomonadota
NI-03	*Acinetobacter pittii* (98%)	MH223594.1	First-instar nymph	Pseudomonadota
NI-04	*Moraxella osloensis* (98%)	MH223595.1	First-instar nymph	Pseudomonadota
PA-01	*Methylobacterium hispanicum* (99%)	PV888580	Pre-adult female	Pseudomonadota
PA-04	*Microbacterium marinilacus* (90%)	PV888581	Pre-adult female	Actinobacteriota
PA-05	*Bacillus* sp. (97%)	PV888582	Pre-adult female	Bacillota
AD-03	*Bacillus* sp. (94%)	PV888584	Adult female	Bacillota
AD-04	*Microbacterium marinilacus* (93%)	PV888583	Adult female	Actinobacteriota

^a^ Similarity percentage was estimated by considering the number of nucleotide substitutions between a pair of sequences divided by the total number of compared bases × 100%.

**Table 4 microorganisms-13-01930-t004:** Fatty acid profiles detected in different developmental stages of the lac insect *L. axin axin*.

Fatty Acid	Eggs	First-Instar Nymphs	Pre-Adult Females	Adult Females	Axe Wax
	Relative Abundance (%)				
Palmitic (C16:0)	2.918	21.78	4.53	4.15	0.389
Stearic (C18:0)	43.82	31.90	43.08	49.68	40.99
Oleic (C18:1)	13.74	8.52	21.62	15.22	17.41
Linoleic (C18:2)	16.14	13.52	19.20	16.08	19.79
Arachidic (C20:0)	17.97	5.42	10.84	13.29	13.66
Behenic (C22:0)	2.32	3.22	ND	ND	0.561
Butyric (C4:0)	ND ^a^	ND	ND	ND	0.258
Valeric (C5:0)	ND	ND	0.157	0.277	ND
Undecanoic	ND	ND	ND	ND	6.924

^a^ ND = Not determined.

## Data Availability

The original contributions presented in this study are included in the article/Appendix A. Further inquiries can be directed to the corresponding author.

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
