# Peer review of "Analysis of the Bacterial Community and Fatty Acid Composition in the Bacteriome of the Lac Insect Llaveia axin axin"

_microorganisms, 2025, doi:10.3390/microorganisms13081930_

Round 1
Reviewer 1 Report
Comments and Suggestions for Authors
The manuscript by Rincón-Rosales et al. explores the contribution of bacterial communities to the biology of the lac insect Llaveia axin axin. At first glance, the study appears methodologically sound, and I found the description of the neuro-fuzzy controlled biosystem for insect rearing particularly interesting from a microbiologist's perspective. However, upon closer examination of the results and their interpretation, several critical inconsistencies and ambiguities arise, which must be addressed before the manuscript can be considered for publication.
- Line 140: It remains unclear whether DNA was actually extracted from egg samples. According to the results section, it appears that it was, so this should be clarified.
- Line 257: The term metagenomic DNA is misleading here. The study performed high-throughput sequencing of 16S rRNA gene fragments, not true metagenomics.
- Line 261: There should be a reference to Table 1 here, and the table itself should be included. Additionally, the description of microbial diversity indices would be more appropriate in this section, before the taxonomic breakdown. Please reorganize accordingly and place the table after the relevant text.
- Throughout the text, Bacteroidetes and Actinobacteria should be updated to Bacteroidota and Actinobacteriota in accordance with current taxonomy. The figures already reflect this correctly.
- Figure 4: The figure is supposed to show microbial composition across insect developmental stages, including eggs, which are discussed in the text, but egg-stage data are missing from the figure. It would also be clearer if the order of bars in the stacked chart followed the natural order of development (top = early, bottom = late stage).
- Lines 257-269: There is a clear mismatch between the text and Figure 4. For example, in the nymph stage, Pseudomonadota comprises ~40% of the community, not 90% as stated. Acidobacteriota, which make up a significant portion (~40%), are not mentioned at all. This discrepancy must be corrected.
- Figure 5: Again, the figure and text are not aligned. The text describes four developmental stages, but the figure only shows three, and not in proper order. The genus Sodalis, which is emphasized in the text (lines 274-288), appears only as a minor component in the figure, and the rest of the genera also appear disordered. When mentioning dominant genera in the text, it would be helpful to include the phylum in parentheses for clarity.
- Lines 324-329: Calculating microbial diversity indices based on colony-forming units from selective media is problematic. These results reflect only the subset of microbes capable of growing on the chosen media. Many environmentally relevant taxa (e.g., Acidobacteriota) are known oligotrophs and likely wouldn’t grow under such rich conditions. Furthermore, not all bacteria tolerate agar as a gelling agent, alternatives like phytagel or xanthan gum might be better for future attempts. Given these limitations, the diversity indices derived from cultivation are not meaningful and should be removed or heavily qualified.
- Section 3.4: It would be interesting to assess the relative abundance of the cultivated isolates within the full sequencing dataset of Illumina 16S rRNA gene fragments. One possible approach: install local BLAST, create a custom database from your 16S reads, and BLAST your isolate sequences against it. Alternatively, extract dominant OTUs using QIIME2 and BLAST your isolates online against those to determine whether you successfully recovered any major community members or not.
- The terms meta-phylogenetic and meta-taxonomic are unusual and potentially confusing. Typically, such analyses are referred to simply as phylogenetic or taxonomic when referring to amplicon-based community profiling. Please revise or clarify what is meant.
- The Discussion section contains very limited interpretation of what roles the detected bacterial taxa (both cultured and uncultured) may play in symbiosis with Llaveia axin axin. This should be expanded, ideally with reference to existing literature on insect-microbe interactions.
- While a BioProject number is provided in the manuscript, no sequence data appear to have been uploaded to GenBank. Please ensure that raw sequencing reads are deposited and accessible.
Author Response
Response to the reviewer 1 comments
The manuscript by Rincón-Rosales et al. explores the contribution of bacterial communities to the biology of the lac insect Llaveia axin axin. At first glance, the study appears methodologically sound, and I found the description of the neuro-fuzzy controlled biosystem for insect rearing particularly interesting from a microbiologist's perspective. However, upon closer examination of the results and their interpretation, several critical inconsistencies and ambiguities arise, which must be addressed before the manuscript can be considered for publication.
Response: Thank you very much for your thoughtful and constructive comments. We appreciate your interest in our work, particularly the neuro-fuzzy controlled biosystem approach. We fully understand your concerns regarding the interpretation of the results and have taken great care to revise the manuscript accordingly. We've clarified the points you raised to ensure the data are presented more clearly and consistently. We hope the updated version addresses your concerns and strengthens the overall quality of the study.
1. Line 140: It remains unclear whether DNA was actually extracted from egg samples. According to the results section, it appears that it was, so this should be clarified.
Response: In line 140, we have replaced the word excluding with including to accurately reflect that DNA was also extracted from the egg samples.
2. Line 257: The term metagenomic DNA is misleading here. The study performed high-throughput sequencing of 16S rRNA gene fragments, not true metagenomics.
Response: In line 257, we modified the text to read: “High-throughput sequencing of 16S rRNA gene amplicons obtained from total DNA,” in order to avoid confusion regarding the methodology used in the study.
3. Line 261: There should be a reference to Table 1 here, and the table itself should be included. Additionally, the description of microbial diversity indices would be more appropriate in this section, before the taxonomic breakdown. Please reorganize accordingly and place the table after the relevant text.
Response: Thank you for the observation. We reorganize the information; we place the table after the corresponding text.
4. Throughout the text, Bacteroidetes and Actinobacteria should be updated to Bacteroidota and Actinobacteriota in accordance with current taxonomy. The figures already reflect this correctly.
Response: Thank you for the observation. We have corrected the observations.
5. Figure 4: The figure is supposed to show microbial composition across insect developmental stages, including eggs, which are discussed in the text, but egg-stage data are missing from the figure. It would also be clearer if the order of bars in the stacked chart followed the natural order of development (top = early, bottom = late stage).
Response: Thank you for the observation. We have corrected the figure 4 followed the natural order of development.
6. Lines 257-269: There is a clear mismatch between the text and Figure 4. For example, in the nymph stage, Pseudomonadota comprises ~40% of the community, not 90% as stated. Acidobacteriota, which make up a significant portion (~40%), are not mentioned at all. This discrepancy must be corrected.
Response: Thank you for the observation. We replace the paragraph of line 263-270 with the following:
Taxonomic classification at the phylum level (Figure 4) revealed a marked ontogenetic restructuring of the bacterial community associated with Llaveia axin axin. The nymphal stage exhibited the highest phylum-level diversity, with detectable proportions of Verrucomicrobiota, Actinomycetota, Acidobacteriota, Bacteroidota, Chloroflexota, and Pseudomonadota. Among these, Pseudomonadota and Acidobacteriota were notably abundant, suggesting a complex and metabolically versatile microbial community during this early stage.
In contrast, pre-adult and adult females showed a striking simplification of the bacteriome, with near-complete dominance by Bacteroidota. This transition may reflect developmental constraints or selection for specialized endosymbionts optimized for mature host physiology. The consistent dominance of Bacteroidota in later stages underscores their likely importance in the adult insect’s metabolic and symbiotic functioning.
These patterns demonstrate a clear phylum-level succession during development, from a phylogenetically diverse and potentially environmentally acquired microbiota in nymphs, toward a highly specialized and reduced symbiotic community in adult stages.
7. Figure 5: Again, the figure and text are not aligned. The text describes four developmental stages, but the figure only shows three, and not in proper order. The genus Sodalis, which is emphasized in the text (lines 274-288), appears only as a minor component in the figure, and the rest of the genera also appear disordered. When mentioning dominant genera in the text, it would be helpful to include the phylum in parentheses for clarity.
Response: Thank you for the comments. Upon closer examination of Figure 5, we confirm that our original description was inaccurate in some respects.
To correct what is shown in Figure 5, we now emphasize the high genus-level diversity observed in nymphs and the near-exclusive dominance of Sodalis in both pre-adult and adult stages. The corrected paragraph now reads:
16S rRNA gene-based metagenomic profiling revealed distinct shifts in the relative abundance and complexity of bacterial genera across the developmental stages of Llaveia axin axin. In first-instar nymphs, a highly diverse bacterial community was detected, including notable genera such as Bradyrhizobium (Pseudomonadota), Burkholderia (Pseudomonadota), Koribacter (Acidobacteriota), Solibacter (Acidobacteriota), Immundisolibacter (Actinobacteriota), Walczuchella (Bacteroidota), and Sodalis (Pseudomonadota), along with several unclassified taxa (e.g., FJ479568_g, GU127739_g, PAC000121_g). In pre-adult and adult females, the community structure shifted towards near-monodominance by the genus Sodalis (Pseudomonadota). This pattern suggests a stage-dependent restructuring and simplification of the bacteriome during maturation.
8. Lines 324-329: Calculating microbial diversity indices based on colony-forming units from selective media is problematic. These results reflect only the subset of microbes capable of growing on the chosen media. Many environmentally relevant taxa (e.g., Acidobacteriota) are known oligotrophs and likely wouldn’t grow under such rich conditions. Furthermore, not all bacteria tolerate agar as a gelling agent, alternatives like phytagel or xanthan gum might be better for future attempts. Given these limitations, the diversity indices derived from cultivation are not meaningful and should be removed or heavily qualified.
Response: Thank you for this thoughtful observation. As the reviewer correctly points out, oligotrophic and fastidious organisms may be underrepresented or absent in these conditions, and alternative gelling agents such as phytagel or xanthan gum may indeed improve future cultivation efforts.
However, the use of culture-dependent approaches to estimate microbial diversity, while inherently limited, is still commonly applied in insect symbiosis studies—particularly when aiming to recover viable isolates for downstream analyses such as biochemical characterization, functional assays, or symbiont-host interaction studies. In our research group and in several related studies, CFU-based indices have provided preliminary insights into culturable diversity, especially in insect-associated microbiomes.
We have noted that future work will consider more refined cultivation methods and complementary culture-independent analyses to better capture total microbial diversity.
9. Section 3.4: It would be interesting to assess the relative abundance of the cultivated isolates within the full sequencing dataset of Illumina 16S rRNA gene fragments. One possible approach: install local BLAST, create a custom database from your 16S reads, and BLAST your isolate sequences against it. Alternatively, extract dominant OTUs using QIIME2 and BLAST your isolates online against those to determine whether you successfully recovered any major community members or not.
Response: Thank you for the observation. In our case, the relative abundance was determined for the bacterial isolates obtained from the insect using the ARDRA genomic fingerprints and in this way, it was possible to calculate the abundance for each stage of the insect, data that are found in Table 1 and Table 2. However, in the analysis of the diversity of the bacterial communities (independent culture method) the richness of the bacterial species that make up the community was also determined through the Shannon and Weaver index.
10. The terms meta-phylogenetic and meta-taxonomic are unusual and potentially confusing. Typically, such analyses are referred to simply as phylogenetic or taxonomic when referring to amplicon-based community profiling. Please revise or clarify what is meant.
Response: Thank you for the observation. We've modified the term.
11. The Discussion section contains very limited interpretation of what roles the detected bacterial taxa (both cultured and uncultured) may play in symbiosis with Llaveia axin axin. This should be expanded, ideally with reference to existing literature on insect-microbe interactions.
Response: Thank you very much for the information. We have expanded the information of the discussion on what role detected bacterial taxa (both cultured and uncultured) can play in the symbiosis with Llaveia axin axin:
At the functional level, our results allow us to propose plausible symbiotic roles for the major bacterial taxa detected in Llaveia axin axin. For example, the early dominance of Sodalis suggests its involvement in nutritional provisioning, consistent with its role as a facultative endosymbiont in other Hemiptera and Coleoptera, where it contributes to the biosynthesis of essential amino acids and cofactors during critical developmental stages (Oakeson et al., 2014).
Consistent with previous reports, Walczuchella monophlebidarum, described by Rosas-Pérez et al. (2014) as the primary symbiont of L. axin axin, was consistently detected throughout all developmental stages and was particularly dominant in adult females. This supports its status as an obligate endosymbiont with a highly reduced genome, specialized in essential amino acid biosynthesis, a typical feature of primary symbionts in scale insects.
In addition, the detection of Blattabacteriaceae and Enterobacteriaceae in later developmental stages suggests potential roles in nitrogen recycling, B-vitamin biosynthesis, and hydrocarbon metabolism. These functions are well-documented in symbionts of cockroaches, aphids, and mealybugs (Sabree et al., 2009) (Hansen & Moran, 2011).
Among the culturable taxa, we recovered genera such as Pseudomonas, Bacillus, and Microbacterium, which are known for their antimicrobial properties, roles in cuticle conditioning, and support of digestive functions in other phytophagous insects. These bacterial genera have also been implicated in plant compound degradation and host immune modulation (Ruiu, 2015), (Ugwu et al., 2020.
Together, these findings provide a more comprehensive ecological and functional framework for interpreting the microbial community associated with L. axin axin, integrating both culture-dependent and culture-independent data.
12. While a BioProject number is provided in the manuscript, no sequence data appear to have been uploaded to GenBank. Please ensure that raw sequencing reads are deposited and accessible.
Response: Thank you for the observation. We already make sure that the raw sequences are deposited and accessible.
Reviewer 2 Report
Comments and Suggestions for Authors
This paper analyzed the microbiome of the bacteriome of the lac insect, and economically important insect. It combines metabarcoding with fatty acid composition.
Major comments:
It looks like this system has been studied before. You cited reference 8 in line 64, but that was a review paper, so it is more appropriate to cite the paper cited by reference 8: Rosas-Pérez et al. 2014, DOI 10.1093/gbe/evu049 . It also is not right to say "a Flavobacterium and an Enterobacterium are present" when you have more accurate information; namely, the full species name of the Flavobacterium Walczuchella monophlebidarum. You should go into more detail about what is already known about this species' microbiome. Re-arrange the introduction to go from broad concepts to specifics, ending with an introduction of what is known about L. axin axin's microbiome. You also need to cite the previous work in the discussion: Walczuchella appears to dominate according to Figure 5, so it's odd that you didn't state this once in the discussion.
I do not know what a "neuro-fuzzy controlled biosystem" is, or why it is important enough to mention in the abstract. It is not sufficient for the abstract to say the insects were reared? In the methods, you mention this biosystem but do not explain what it actually is in enough detail for someone to replicate it, nor do you cite a single source. If this biosystem is brand new to science, then publish a separate paper on it, or at least add enough details to the methods to describe how someone could perfectly replicate your set-up [you may also want to add the existance of a novel rearing system to the title]. If your rearing system has been described before, either in your own work or in someone elses, then cite that paper here.
Again in lines, if this is a novel system, then you must add much, much, much more data to the methods on how you created it, with lists of machines and their brands and providers, the programs used and a link to the code you wrote on GitHub or some other depository, etc., and then modify the title and abstract to make it clear that the invention of this system is part of the paper. You may want to write a separate paper if you invented neuro-fuzzy breeding systems. In any casem cut out claims of the system providing "valuable biological data." I personally see zero evidence for any data of any kind provided by your rearing system. Your rearing system worked. That is not the same as providing data! I see no evidence that this platform is "robust" because you provide no data comparing it to other platforms. I see no evidence that it advanced "microbiological, biochemical, and conservation research." Delete all of this. All you did was rear an insect from egg to adult, which may or may not be a novel achievement for this species [you provided zero references to prior rearing efforts], but that's it.
Endosymbionts are rarely if ever culturable on standard media, and usually require insect cell lines for culturing. I suspect most if not all of the microbes cultured on the PY-Ca and YEM media are contaminants, not symbionts. Can you argue otherwise? Can you present evidence from other insects that culturable, aerobic, free-living microbes exist fully inside the anaerobic cells of the bacteriocyte?
279-283 What you wrote here does not match what I see in Figure 5. Something is wrong, and I cannot trust your results.
I think you can delete the phylum-level deta and figure 4 completely from the paper. Phylum-level data is almost always worthless, especially if you have genus level-data. Imagine you are describing the contents of a lake. If you present a list of the genera of fish, amphibians, reptiles, and birds in the lake, then why bother also saying the lake has phylum Chordata?
Minor comments:
43 and 453 add comma before "but also"
52 Axe or AXE?
121 What instar or days after hatching were "pre-adult females?" Also, how did you ensure there were no males? [If the species is parthenogenic, say so.]
128 Where are the bacteriomes located? Do they wrap around part of the gut, and if so, which part? Or do they connect to the ovaries? Provide more detailed information such that someone could replicate this research based on the methods described.
249-250 Nobody ever doubted that this species does incomplete metamorphosis. No need to confirm the obvious.
Figure 4+5: Where are the eggs? And why are the bars not in age order from nymph to pre-adult to adult?
338-339 First, this is discussion text, not results. Second, the majority of bacteria are non-culturable. Lack of culturable bacteria does not mean they were inaccessible in crypts, especially since crushing the eggs would eman access to crypts. It just means the symbionts are not-culturable, period. Third, eggs don't have crypts. You are thinking of the crypts for symbionts found in the guts of certain insects, but that has nothing to do with eggs, and those are typically for facultative, free-living symbonts like Burkholderia in stinkbugs. Please rewrite or delete.
363-375 italics needed
399-405 Delete. It's redundnat with the introduction.
406-411 Delete unless you provide several pages more data on how to build this system. If the methods are not enough for someone reading this paper to build an identical system, then this paper is not the place to discuss the merits of a fuzzy system vs others.
412-420 Us genera instead of phyla whenever posible.
425 I'd like more detailed descriptions of what was found in other insects. "host-symbiont specificity is developmentally regulated" is too vague. In another species, was there specific evidence that a specific genus of endosymbiont was replaced by another during development? if so, provide the species of insect and species or genera of bacteria for each specific example and cite them seperately.
426-435 It is not clear what you learned that was different between ARDRA and 16S profiling. In 431, are these cultivable only sequences or the full bacteriocyte metabarcode?
440-441 "implies a regulated lipid biosynthesis pathway tightly linked to development" I see no evidence of regulation or a linkage, and certainly not the tightness of the linkage. State only that the fatty accids different in different life stages.
444-445 There's a grammar error here that causes contradiction: you say the fatty acids are "exclusively in AXE wax," meaning they are not found in bacteria or insect cuticle, yet then claim the bacteria are involved in AXE production and cuticle.
447-448 Give specific examples: which species of insect and which species of symbiont. Are any of these wax producers?
456-458 Delete
Author Response
Reviewer 2
This paper analyzed the microbiome of the bacteriome of the lac insect, and economically important insect. It combines metabarcoding with fatty acid composition.
Major comments:
It looks like this system has been studied before. You cited reference 8 in line 64, but that was a review paper, so it is more appropriate to cite the paper cited by reference 8: Rosas-Pérez et al. 2014, DOI 10.1093/gbe/evu049 . It also is not right to say "a Flavobacterium and an Enterobacterium are present" when you have more accurate informatio1n; namely, the full species name of the Flavobacterium Walczuchella monophlebidarum. You should go into more detail about what is already known about this species' microbiome. Re-arrange the introduction to go from broad concepts to specifics, ending with an introduction of what is known about L. axin axin's microbiome. You also need to cite the previous work in the discussion: Walczuchella appears to dominate according to Figure 5, so it's odd that you didn't state this once in the discussion.
Response: We thank the comment. We agree that citing the primary source, Rosas-Pérez et al. (2014), is more appropriate than referencing a review article. Accordingly, we have revised the manuscript to cite this study directly in the Introduction and Discussion.
In addition, we have corrected the terminology regarding the symbionts of Llaveia axin axin. Rather than using the general terms "a Flavobacterium and an Enterobacterium," we now specify Walczuchella monophlebidarum as the Flavobacterium endosymbiont, as originally described by Rosas-Pérez et al. (2014). We have also revised the Introduction to include in line 61 more detail about what is already known about the microbiome of L. axin axin, including the dominance and functional significance of Walczuchella:
Symbiotic associations between insects and microorganisms play essential roles in the host’s nutrition, development, and ecological adaptation. Among Hemiptera, endosymbionts are particularly important in sap-feeding lineages, where symbiotic bacteria complement deficient diets by synthesizing essential nutrients. In members of the superfamily Coccoidea (scale insects), symbiotic systems are diverse and often include specialized bacteriomes housing obligate endosymbionts. Within this group, the Monophlebidae family is known to host complex microbial communities, including both obligate and facultative symbionts. One such species is Llaveia axin axin, a phloem-feeding giant scale insect endemic to Mesoamerica. Previous studies have shown that this insect harbors two main bacterial symbionts: Walczuchella monophlebidarum (Flavobacteriaceae), an obligate intracellular bacterium residing in the bacteriocytes, and an Enterobacteriaceae species potentially involved in nitrogen recycling and vitamin synthesis (Rosas-Pérez et al., 2014).
However, the full extent of bacterial diversity associated with L. axin axin across its developmental stages remains unexplored. In this study, we employed high-throughput 16S rRNA gene sequencing to characterize the structure and composition of bacterial communities inhabiting the bacteriomes of L. axin axin at different stages of development, expanding our understanding of its microbiome dynamics.
In Discussion, the findings of Figure 5 on Walczuchella monophlebidarum were integrated and reference to Rosas-Pérez et al. (2014), as requested by the reviewer:
Our results revealed a clear dominance of a Flavobacteriaceae taxon, particularly evident in adult stages (Figure 5). Based on previous research, this bacterium has been identified as Walczuchella monophlebidarum, an obligate endosymbiont of Llaveia axin axin, localized within specialized bacteriocytes and vertically transmitted across generations (Rosas-Pérez et al., 2014). This supports the interpretation that W. monophlebidarum is the primary symbiotic partner in this species, consistent with the dominant OTU observed in our meta-phylogenetic data.
The secondary symbiont, previously characterized as a member of the Enterobacteriaceae, was also detected across developmental stages but at much lower abundance. This pattern is consistent with its proposed facultative role in nutritional complementation or metabolic flexibility, as suggested in other scale insects (Rosas-Pérez et al., 2014).
By confirming the presence and dominance of W. monophlebidarum, our data validate and expand on existing knowledge of the L. axin axin microbiome. Moreover, the stage-dependent fluctuation in community structure suggests dynamic regulation of bacterial symbionts during host development, a pattern observed in other insect-symbiont systems as well.
I do not know what a "neuro-fuzzy controlled biosystem" is, or why it is important enough to mention in the abstract. It is not sufficient for the abstract to say the insects were reared? In the methods, you mention this biosystem but do not explain what it actually is in enough detail for someone to replicate it, nor do you cite a single source. If this biosystem is brand new to science, then publish a separate paper on it, or at least add enough details to the methods to describe how someone could perfectly replicate your set-up [you may also want to add the existance of a novel rearing system to the title]. If your rearing system has been described before, either in your own work or in someone elses, then cite that paper here.
Response: In the Abstract section, in line 20 the phrase "neuro-fuzzy controlled biosystem" was deleted and changed to "at the laboratory level, in controlled conditions". And in the methodology, the reference Morales-Mancilla et al., 2015, was added for a better understanding of the functioning of the biosystem.
Likewise, we want to clarify that the neuro-fuzzy controlled biosystem allowed to establish the optimal conditions to achieve the breeding of this insect, since in natural conditions it is very difficult to obtain them, since it is in danger of extinction.
Again in lines, if this is a novel system, then you must add much, much, much more data to the methods on how you created it, with lists of machines and their brands and providers, the programs used and a link to the code you wrote on GitHub or some other depository, etc., and then modify the title and abstract to make it clear that the invention of this system is part of the paper. You may want to write a separate paper if you invented neuro-fuzzy breeding systems. In any casem cut out claims of the system providing "valuable biological data." I personally see zero evidence for any data of any kind provided by your rearing system. Your rearing system worked. That is not the same as providing data! I see no evidence that this platform is "robust" because you provide no data comparing it to other platforms. I see no evidence that it advanced "microbiological, biochemical, and conservation research." Delete all of this. All you did was rear an insect from egg to adult, which may or may not be a novel achievement for this species [you provided zero references to prior rearing efforts], but that's it.
Response: Thank you for this detailed and constructive feedback. We appreciate your related to the rearing system described in our manuscript. In response, we have now included a citation to previously published work [Morales-Mancilla et al., 2015] reporting the application or testing of this neuro-fuzzy controlled biosystem rearing approach.
Endosymbionts are rarely if ever culturable on standard media, and usually require insect cell lines for culturing. I suspect most if not all of the microbes cultured on the PY-Ca and YEM media are contaminants, not symbionts. Can you argue otherwise? Can you present evidence from other insects that culturable, aerobic, free-living microbes exist fully inside the anaerobic cells of the bacteriocyte?
Response: Thank you for this important observation. We agree that many obligate endosymbionts, particularly those residing within bacteriocytes, are unculturable under standard laboratory conditions. Our study did not aim to isolate obligate intracellular symbionts but rather to characterize the culturable fraction of the microbial community associated with the bacteriome. We acknowledge that some of the isolates obtained on PY-Ca and YEM media are likely to be environmental or transient microbes, possibly external or gut-associated bacteria, rather than true intracellular symbionts.
However, we have revised the manuscript to clarify that the culturable bacteria we recovered are not assumed to be obligate intracellular symbionts, but rather represent a subset of facultative, potentially transient, or environmentally acquired bacteria associated with bacteriome tissue. This is consistent with previous reports where facultative symbionts like Serratia symbiotica and Hamiltonella defensa have been successfully cultured from aphid hosts (Sabri et al., 2011). Similarly, culturable Burkholderia spp. have been isolated from midgut crypts in stinkbugs (Kikuchi et al., 2007), demonstrating that some insect-associated symbionts are culturable under appropriate conditions.
Based on this we have now included these references and two paragraph in Discussion, to clarify that the isolates obtained represent culturable, potentially associated bacteria, but not necessarily obligate endosymbionts:
It is important to note that the bacteria isolated on PY-Ca and YEM media do not necessarily represent obligate intracellular endosymbionts, which are typically unculturable under standard aerobic conditions due to their strict dependence on host cells and reduced genomes. Rather, our isolates likely correspond to facultative or transiently associated bacteria, some of which may be extracellular symbionts or environmental microbes colonizing the bacteriome region.
In other insect systems, culturable facultative symbionts have been recovered from tissues associated with bacteriomes or hemolymph. For instance, Serratia symbiotica and Hamiltonella defensa, both facultative symbionts of aphids, have been successfully cultured on cell-free media under specific conditions (Sabri et al., 2011). Additionally, studies in stinkbugs have shown that culturable Burkholderia spp. symbionts colonize specialized midgut crypts and can be transmitted environmentally (Kikuchi et al., 2007). Thus, while most obligate endosymbionts of bacteriocytes (e.g., Buchnera aphidicola, Carsonella ruddii) are indeed unculturable (Moran et al., 2008), the bacterial communities associated with the bacteriome may include culturable, aerobic microorganisms of potential ecological relevance.
279-283 What you wrote here does not match what I see in Figure 5. Something is wrong, and I cannot trust your results.
Response: Thank you for your review. Upon closer examination of Figure 5, we confirm that our original description was inaccurate in some respects.
To correct what is shown in Figure 5, we now emphasize the high genus-level diversity observed in nymphs and the near-exclusive dominance of Sodalis in both pre-adult and adult stages. The corrected paragraph now reads:
16S rRNA gene-based metagenomic profiling revealed distinct shifts in the relative abundance and complexity of bacterial genera across the developmental stages of Llaveia axin axin. In first-instar nymphs, a highly diverse bacterial community was detected, including notable genera such as Bradyrhizobium (Pseudomonadota), Burkholderia (Pseudomonadota), Koribacter (Acidobacteriota), Solibacter (Acidobacteriota), Immundisolibacter (Actinobacteriota), Walczuchella (Bacteroidota), and Sodalis (Pseudomonadota), along with several unclassified taxa (e.g., FJ479568_g, GU127739_g, PAC000121_g). In pre-adult and adult females, the community structure shifted towards near-monodominance by the genus Sodalis (Pseudomonadota). This pattern suggests a stage-dependent restructuring and simplification of the bacteriome during maturation.
I think you can delete the phylum-level data and figure 4 completely from the paper. Phylum-level data is almost always worthless, especially if you have genus level-data. Imagine you are describing the contents of a lake. If you present a list of the genera of fish, amphibians, reptiles, and birds in the lake, then why bother also saying the lake has phylum Chordata?
Response: We appreciate your comment regarding Figure 4. After reviewing the figure carefully, we acknowledge that the original text describing phylum-level results was inaccurate and not aligned with the visual data. We have corrected the description to reflect the actual patterns shown in the figure.
However, we believe that Figure 4 provides essential context for understanding the phylogenetic transitions in the bacteriome of Llaveia axin axin during development. It highlights a shift from a diverse microbial community in the nymphal stage to a highly specialized Bacteroidota-dominated community in adult stages, which is not fully captured by genus-level analysis alone (Figure 5). Therefore, we propose to retain Figure 4 and replace the paragraph of line 263-270 with the following:
Taxonomic classification at the phylum level (Figure 4) revealed a marked ontogenetic restructuring of the bacterial community associated with Llaveia axin axin. The nymphal stage exhibited the highest phylum-level diversity, with detectable proportions of Verrucomicrobiota, Actinomycetota, Acidobacteriota, Bacteroidota, Chloroflexota, and Pseudomonadota. Among these, Pseudomonadota and Acidobacteriota were notably abundant, suggesting a complex and metabolically versatile microbial community during this early stage.
In contrast, pre-adult and adult females showed a striking simplification of the bacteriome, with near-complete dominance by Bacteroidota. This transition may reflect developmental constraints or selection for specialized endosymbionts optimized for mature host physiology. The consistent dominance of Bacteroidota in later stages underscores their likely importance in the adult insect’s metabolic and symbiotic functioning.
These patterns demonstrate a clear phylum-level succession during development, from a phylogenetically diverse and potentially environmentally acquired microbiota in nymphs, toward a highly specialized and reduced symbiotic community in adult stages.
Minor comments:
43 and 453 add comma before "but also"
Response: Thank you for the observation. We've already resolved it.
52 Axe or AXE?
Response: Thank you for the observation. "Axe" is the traditional name of the resin and not an acronym, so the correct way to write it is "Axe", not "AXE".
121 What instar or days after hatching were "pre-adult females?" Also, how did you ensure there were no males? [If the species is parthenogenic, say so.]
Response: Thank you for your valuable question. In our study, “pre-adult females” refers to individuals at the third instar, approximately 90–100 days after hatching, just before the transition to the mature, sessile adult stage that secretes the lac resin. These individuals were identified based on morphological features such as body enlargement, increased pigmentation, partial secretion of wax, and transition to arboreal behavior.
Although males of Llaveia axin axin have been reported in the literature, they are extremely rare and are typically absent in laboratory-reared populations. Throughout our neuro-fuzzy controlled rearing process, no male individuals were observed, and all collected individuals developed into morphologically typical females. However, parthenogenesis has not been confirmed in this species, and the absence of males under controlled conditions is interpreted as a result of environmental or genetic sex-bias mechanisms rather than obligate parthenogeny. We have now clarified this point in the revised manuscript:
A neuro-fuzzy controlled biosystem was implemented to rear Llaveia axin axin under stable laboratory conditions (Figure 1). To simulate the insect's natural habitat, the system integrated neural network training with fuzzy logic-based real-time regulation, adjusting environmental parameters such as temperature, relative humidity, and soil moisture in response to developmental stage. Jatropha curcas seedlings were cultivated and acclimated for seven days before carefully placing freshly laid eggs of L. axin axin at the base of each plant. The complete life cycle—from hatching to adult emergence—was monitored under these dynamic conditions. Specimens were collected at key timepoints for genomic and biochemical analyses. Individuals designated as “pre-adult females” corresponded to the third-instar stage, occurring approximately 90–100 days post-hatching, just before attaining full maturity. Identification was based on morphological traits such as increased body size, partial wax secretion, and transition to sessile behavior on host stems, in agreement with the ontogenetic stages depicted
128 Where are the bacteriomes located? Do they wrap around part of the gut, and if so, which part? Or do they connect to the ovaries? Provide more detailed information such that someone could replicate this research based on the methods described.
Response: Thank you for your comment. In Llaveia axin axin, bacteriomes are discrete, whitish organs located bilaterally in the posterior abdominal cavity, adjacent to the ovaries and the hindgut. Based on anatomical dissection under stereomicroscopy, they do not wrap around the gut, but are clearly associated with the reproductive system, suggesting possible transovarial transmission of endosymbionts. Their position and pigmentation made them distinguishable from fat bodies and other internal tissues, allowing precise excision for downstream metagenomic analysis.
We have now included a more detailed anatomical description of bacteriome location and excision procedure in the Materials and Methods section to improve reproducibility, the paragraph of line 120-133 now reads:
Samples were collected from different developmental stages of L. axin axin, including eggs (E), first-instar nymphs (FIN), pre-adult females (PAF), and adult females (AF). Each specimen was carefully cleaned to remove the white wax accumulated on the cuticle, fol-lowed by surface disinfection through immersion in 70 % ethanol for 10 min, and subse-quently washed multiple times with sterile phosphate-buffered saline (PBS) to eliminate any residual contaminants. Under sterile conditions, individual insects were dissected using fine sterile forceps under a stereoscopic microscope to extract the targeted tissues. Dissection of L. axin axin females was conducted under a stereoscopic microscope to extract the targeted tissues. Bacteriomes were identified as paired, ovoid, whitish organs located in the posterior abdominal cavity, positioned laterally to the ovaries and ventrally to the hindgut (Figure 2). These structures were clearly distinct from surrounding tissues due to their color, texture, and bilateral symmetry. Some samples were immediately processed and analyzed after extraction, while others were preserved for subsequent analyses. To preserve sample integrity and prevent oxidative degradation, the extracted materials were thoroughly washed with sterile phosphate-buffered saline (PBS) and sub-sequently stored in 70 % ethanol, following the standardized protocol established by Ramírez-Puebla et al.
249-250 Nobody ever doubted that this species does incomplete metamorphosis. No need to confirm the obvious.
Response: Thank you for the observation, those lines were removed.
Figure 4+5: Where are the eggs? And why are the bars not in age order from nymph to pre-adult to adult?
Response: Thank you for your comment regarding Figures 4 and 5. Wewe have revised both the figures and the accompanying text to reflect only the three developmental stages for which reliable 16S rRNA gene sequencing data were obtained: nymph, pre-adult, and adult.
338-339 First, this is discussion text, not results. Second, the majority of bacteria are non-culturable. Lack of culturable bacteria does not mean they were inaccessible in crypts, especially since crushing the eggs would eman access to crypts. It just means the symbionts are not-culturable, period. Third, eggs don't have crypts. You are thinking of the crypts for symbionts found in the guts of certain insects, but that has nothing to do with eggs, and those are typically for facultative, free-living symbonts like Burkholderia in stinkbugs. Please rewrite or delete.
Response: Thank you for the important comment. We decided to delete the text from lines 338-339.
363-375 italics needed
Response: Thank you for the observation. We've already resolved it
399-405 Delete. It's redundnat with the introduction.
Response: Thank you for the observation, those lines were removed.
406-411 Delete unless you provide several pages more data on how to build this system. If the methods are not enough for someone reading this paper to build an identical system, then this paper is not the place to discuss the merits of a fuzzy system vs others.
Response: Thank you for the observation, those lines were removed.
412-420 Use genera instead of phyla whenever possible.
Response: We appreciate the comment. We have corrected the use of gender instead of phyla:
Moreover, the meta-phylogenetic analysis revealed a pronounced ontogenetic shift in the composition of the bacteriome across developmental stages. Genera such as Bradyrhizobium and Burkholderia were prominent in nymphs, whereas Koribacter and Solibacter dominated the bacteriome in pre-adult and adult females. Similar developmental shifts in symbiotic communities have been observed in other hemipterans, including mealybugs and aphids, where early colonization by Gammaproteobacteria gives way to more specialized or diverse bacterial consortia as development progresses. The dramatic reduction in microbial diversity observed in adult females suggests selective maintenance of specific endosymbionts, potentially linked to their sessile behavior and involvement in lac secretion.
425 I'd like more detailed descriptions of what was found in other insects. "host-symbiont specificity is developmentally regulated" is too vague. In another species, was there specific evidence that a specific genus of endosymbiont was replaced by another during development? if so, provide the species of insect and species or genera of bacteria for each specific example and cite them seperately.
Response: Thank you for the observation. We have added insect-specific examples where endosymbiont replacement has been documented during development, including the names of the insects, the bacterial genera involved, and corresponding citations:
At a finer taxonomic resolution, the genus-level analysis indicated early dominance of Sodalis, an endosymbiont frequently associated with nutritional functions, highlighting its critical role in early host development. Its subsequent decline and replacement by Blattabacteriaceae in adult females mirrors patterns observed in other insects where host-symbiont composition changes across developmental stages. For example, in psyllids (Hemiptera: Psylloidea), a shift was observed from Sodalis-like bacteria during juvenile stages to Arsenophonus or other Enterobacteriaceae in adults (Morrow et al., 2017). Similarly, in weevils of the family Dryophthoridae, Nardonella has been replaced in some lineages by Gammaproteobacteria such as Sodalis or Enterobacter across host evolution, with evidence suggesting possible ontogenetic transitions in bacterial load and function (Lefèvre et al., 2004), (Conord et al., 2008). These studies provide clear evidence of taxon-specific shifts in endosymbiont identity linked to developmental or evolutionary dynamics.
426-435 It is not clear what you learned that was different between ARDRA and 16S profiling. In 431, are these cultivable only sequences or the full bacteriocyte metabarcode?
Response: Thank you for the observation. In response, we have inserted the following text on line 435, immediately after reference 48:
Collectively, these two approaches offered complementary insights: ARDRA enabled detection of genotypic patterns among cultivable isolates, while high-throughput 16S profiling captured the full taxonomic structure of the bacterial community, including non-culturable taxa. This integration reinforced the observation of a progressive narrowing of microbial diversity through development.
440-441 "implies a regulated lipid biosynthesis pathway tightly linked to development" I see no evidence of regulation or a linkage, and certainly not the tightness of the linkage. State only that the fatty accids different in different life stages.
Response: Thank you for your observation. We have revised the paragraph accordingly, and it now reads as follows:"
With respect to lipid metabolism, the fatty acid profile across developmental stages highlighted stearic acid (C18:0) as the most abundant fatty acid in all insect stages and in AXE wax. This consistent presence suggests a potential dual role as both a structural and signaling lipid. In contrast, the relative abundance of other fatty acids, such as palmitic (C16:0), linoleic (C18:2), and arachidic acid (C20:0), varied across life stages, indicating developmental stage-specific lipid profiles. Similar patterns have been observed in the white wax scale insect Ericerus pela, where genes related to fatty acid elongation and reduction (e.g., ELOVL, FAR) are associated with wax synthesis.
444-445 There's a grammar error here that causes contradiction: you say the fatty acids are "exclusively in AXE wax," meaning they are not found in bacteria or insect cuticle, yet then claim the bacteria are involved in AXE production and cuticle.
Response: Thank you for pointing out the grammatical inconsistency. We have revised the sentence to clarify that the presence of behenic and undecanoic acids is specifically associated with AXE wax, without implying that these compounds are completely absent in bacterial metabolism or the insect cuticle. The revised text now accurately reflects that bacterial symbionts may influence wax biosynthesis or related cuticle functions, as observed in other insects. This change resolves the contradiction and aligns better with the cited literature:
Notably, the detection of fatty acids such as behenic and undecanoic acid specifically associated with axe wax supports the hypothesis that bacterial metabolic contributions may influence wax secretion and potentially contribute to cuticle-related processes. Comparable findings have been reported in other wax-producing insects, where bacterial symbionts participate in the biosynthesis or structural modification of cuticular hydrocarbons and esters.
447-448 Give specific examples: which species of insect and which species of symbiont. Are any of these wax producers?
Response: Thank you for pointing this out. We agree that the previous phrasing was vague and have now included specific examples. In response, we have revised the discussion to include specific examples of insect species and their associated bacterial symbionts that have been implicated in the synthesis or modification of waxes and cuticular hydrocarbons:
Comparable findings have been reported in other wax-producing insects, such as the honeybee (Apis mellifera), where bacterial symbionts have been associated with modifications in the composition of cuticular waxes, primarily hydrocarbons and wax esters (Cilia et al., 2021). In the case of Ceratitis capitata, gut bacterial symbionts influence cuticular hydrocarbon profiles that are critical for sexual behavior and potentially for hydrocarbon biosynthesis (Moyano et al., 2025). Additionally, in leafcutter ants (Atta sexdens, Atta cephalotes), specific cuticular compounds such as alkyl amides are present only when mutualistic bacteria are associated with the cuticle, suggesting a potential biosynthetic contribution from the symbionts (Fladerer et al., 2023). These cases underscore the potential role of microbial partners in shaping the composition of wax-related compounds in insects.
456-458 Delete
Response: Thank you for the observation, those lines were removed.
Round 2
Reviewer 1 Report
Comments and Suggestions for Authors
The authors have addressed most of the comments. I would just like to clarify one point, if DNA was extracted from the egg stage, could you please explain why this stage is not shown in the bar charts with taxonomic profiling, where only the subsequent three stages appear?
Author Response
Reviewer 1 Comments.
The authors have addressed most of the comments. I would just like to clarify one point, if DNA was extracted from the egg stage, could you please explain why this stage is not shown in the bar charts with taxonomic profiling, where only the subsequent three stages appear?
We thank the reviewer for this clarification. To improve the clarity of the results and ensure consistency with the data presented, we removed the sentences referring to DNA analysis (genomic and metagenomic) from the egg stage and now only present results for the three developmental stages included in the taxonomic profiling bar charts (nymph, pre-adult, and adult).
Thank you very much for this important observation.
Reviewer 2 Report
Comments and Suggestions for Authors
Many of the requested changes have been made. I am satisfied with the revision!
Author Response
Reviewer 2 Comments.
Many of the requested changes have been made. I am satisfied with the revision!
We sincerely thank the reviewer for their valuable and insightful comments, which have greatly contributed to improving our manuscript